# Role and Regulatory Mechanism of circRNA_14820 in the Proliferation and Differentiation of Goat Skeletal Muscle Satellite Cells

**DOI:** 10.3390/ijms25168900

**Published:** 2024-08-15

**Authors:** Pu Yang, Xuelong Li, Chengli Liu, Yanguo Han, Guangxin E, Yongfu Huang

**Affiliations:** College of Animal Science and Technology, Southwest University, Chongqing 400715, China; yp2645642899@163.com (P.Y.); 18793012846@163.com (X.L.); lcl222333@outlook.com (C.L.); hyg2015@swu.edu.cn (Y.H.); eguangxin@126.com (G.E.)

**Keywords:** circ_14820, differentiation, miR-206, proliferation, skeletal muscle satellite cells

## Abstract

Skeletal muscle satellite cells (SMSCs), a type of myogenic stem cell, play a pivotal role in postnatal muscle regeneration and repair in animals. Circular RNAs (circRNAs) are a distinct class of non-coding RNA molecules capable of regulating muscle development by modulating gene expression, acting as microRNAs, or serving as protein decoys. In this study, we identified circ_14820, an exonic transcript derived from adenosine triphosphatase family protein 2 (ATAD2), through initial RNA-Seq analysis. Importantly, overexpression of circ_14820 markedly enhanced the proliferation of goat SMSCs while concomitantly suppressing their differentiation. Moreover, circ_14820 exhibited predominant localization in the cytoplasm of SMSCs. Subsequent small RNA and mRNA sequencing of circ_14820-overexpressing SMSCs systematically elucidated the molecular regulatory mechanisms associated with circ_14820. Our preliminary findings suggest that the circ_14820-miR-206-CCND2 regulatory axis may govern the development of goat SMSCs. These discoveries contribute to a deeper understanding of circRNA-mediated mechanisms in regulating skeletal muscle development, thereby advancing our knowledge of muscle biology.

## 1. Introduction

Skeletal muscle development encompasses the increase in muscle fiber number during embryonic stages and subsequent fiber diameter enlargement postnatally. This intricate process is governed by a complex interplay of transcription factors and signaling pathways. Key gene families, such as myogenic regulatory factor (MRF), myocyte enhancer factor 2 (MEF2), and paired box (Pax), act synergistically to ensure proper muscle development [1,2,3,4]. Additionally, signaling pathways, including Wnt, Notch, PI3K-AKT, and mTOR, are pivotal in regulating myosatellite cell or myoblast development to promote muscle production [5,6,7,8]. Moreover, precise regulation of non-coding RNAs (ncRNAs) is critical in skeletal muscle development [9].

Circular RNA (circRNA), generated by reverse shearing of precursor messenger RNA (pre-mRNA), is a covalently closed circular single-stranded ncRNA molecule [10,11]. Recent studies have shown that circRNAs are heavily localized in the cytoplasm of exosomes or eukaryotic cells, and their expression has significant temporal and tissue specificity [12,13]. Moreover, endogenous circRNAs are characterized by stable structures and sequences, exhibiting extended half-lives throughout evolutionary conservation [14,15,16]. Functionally, circRNAs play pivotal roles in regulating animal growth, metabolism, and disease development. Some circRNAs regulate genes involved in skeletal muscle development via competitive endogenous RNA (ceRNA) mechanisms. For instance, Wei et al. [17] demonstrated that circ-LMO7 regulates bovine myoblast development by adsorbing miR-378-3p to modulate HDAC4 expression. Additionally, Li et al. [18] elucidated the role of the circ-CDR1as-miR-7-IGF1R axis in goat myoblast differentiation. In the cytoplasm, circSGCB acts as a molecular sponge to modulate KLF3 expression, thereby promoting muscle regeneration in ruminants [19].

ATAD2, a member of the adenosine triphosphatase family, exerts control over cell proliferation, apoptosis, and cell fate by modulating various signaling pathways [20]. For instance, ATAD2 deficiency impedes cancer cell proliferation and induces apoptosis through inhibition of the MAPK and PI3K-AKT pathways [21,22]. Furthermore, ATAD2 plays a significant role in regulating cell cycle-related transcription factors like cyclin D2 and cyclin dependent kinase 4 [23]. Due to these functions, ATAD2 has garnered considerable attention in cancer research as a promising target for anticancer therapies and a potential biomarker across various malignant tumors. However, detailed exploration of the specific functions of ATAD2-derived circular transcripts and their involvement in regulating muscle development remains to be elucidated.

Building upon our previous investigation [24] of longissimus dorsi muscle tissues from ET (embryonic day 75) and DC (day 1 postnatal) DaZu black goats, we identified a circular transcript, circ_14820, derived from the ATAD2 gene through high-throughput sequencing. Our focus centered on elucidating the impact of circ_14820 on the proliferation and differentiation of goat skeletal muscle satellite cells (SMSCs), revealing that circ_14820 enhances proliferation while suppressing differentiation of SMSCs. Through comprehensive analysis of small RNA and mRNA sequencing data, we systematically delineated the downstream molecular targets regulated by circ_14820 and constructed a circ_14820-miRNA-mRNA regulatory network. These findings hold significant implications for advancing our understanding of skeletal muscle development. This study aims to provide foundational insights into the mechanisms of circRNA action in goat skeletal muscle development.

## 2. Results

### 2.1. Identification of Circ_14820

Based on the analysis of the circ_14820 sequence obtained from our high-throughput sequencing, it was found to be a circular transcript formed by reverse shearing of exons 8 to 20 of adenosine triphosphatase family protein 2 (ATAD2), with a total length of 1923 nt. We performed a PCR reaction using divergent primers to amplify the circ_14820-containing junction site and performed Sanger sequencing on the product. The results showed that circ_14820 has a reverse shear site (Figure 1A), which is consistent with the high-throughput data. To validate the circular structure of circ_14820, we performed PCR using divergent and convergent primers on cDNA, RNase R-treated cDNA, and genomic DNA (gDNA) isolated from goat dorsal muscle tissue. Agarose gel electrophoresis revealed successful amplification of circ_14820 with divergent primers using cDNA or RNase R-treated cDNA as templates, but not with gDNA. Conversely, efficient amplification was observed with convergent primers using cDNA or gDNA as templates, with no significant bands detected from RNase R-treated cDNA (Figure 1B). Additionally, qPCR analysis demonstrated that circ_14820 exhibited resistance to RNase R nucleic acid exonuclease treatment (Figure 1C), confirming its circular nature. Furthermore, we predicted the secondary structure of circ_14820 using the RNAfold web server (Figure 1D), revealing a minimum thermodynamic free energy of −525.69 kcal/mol.

### 2.2. Spatiotemporal Expression Pattern of Circ_14820

To analyze the spatiotemporal expression pattern of circ_14820, we constructed a tissue expression profile. Our results indicate that circ_14820 exhibits significantly higher expression in various tissues of embryonic 75-day-old goats (ET group) compared to 1-day-old postnatal goats (DC group), with the longissimus dorsi muscle showing the highest expression (Figure 2A). Additionally, circ_14820 expression increases progressively during the proliferation phase of skeletal muscle satellite cells (SMSCs), declines during the initial 3 days of the differentiation phase, and then gradually rises again (Figure 2B). This implies that circ_14820 mainly acts in the tissue development of embryonic goats and likely participates in regulating skeletal muscle development by modulating the proliferation and differentiation of SMSCs.

### 2.3. Circ_14820 Promotes Proliferation of SMSCs

To investigate the impact of circ_14820 on SMSC proliferation, we constructed a lentivirus overexpressing circ_14820 (Figure 3A). As anticipated, circ_14820 expression was markedly elevated in the overexpression group compared to the negative control (Control) (*p* < 0.01) (Figure 3B). Remarkably, overexpression of circ_14820 significantly upregulated transcript levels of the key proliferation markers PAX7, CCND1, and CCND2 in SMSCs (*p* < 0.01) (Figure 3C). PAX7 protein expression followed the same trend as its mRNA expression (Figure 3D). Furthermore, CCK-8 assays conducted 24 h post-circ_14820 overexpression revealed significantly higher cellular absorbance values compared to the control group (*p* < 0.05) (Figure 3E). Cell cycle analysis indicated a notable decrease in the G0/G1 phase cell population and a substantial increase in the S-phase cell population following circ_14820 overexpression (*p* < 0.01) (Figure 3F,G). Additionally, EdU assays showed a significant increase in the number of newly formed nuclei in response to circ_14820 (*p* < 0.01) (Figure 3H). These results suggest that circ_14820 may positively affect the proliferation of SMSCs.

### 2.4. Circ_14820 Inhibits Differentiation of SMSCs

To investigate the influence of circ_14820 on the differentiation of goat skeletal muscle satellite cells (SMSCs), cells were induced with differentiation medium following transfection. Our results demonstrate that circ_14820 overexpression markedly suppressed the expression of key differentiation marker genes, including MyHC, MyoG, MyoD, and Myf5, in SMSCs (Figure 4A). Consistently, protein levels of MyHC and MyoD mirrored the trends observed in mRNA expression (Figure 4B). Furthermore, immunofluorescence analysis revealed a significant reduction in myotube formation upon circ_14820 overexpression (Figure 4C).

### 2.5. Nucleoplasmic Localization of Circ_14820

We assessed the subcellular localization of circ_14820 in cells using nucleoplasmic separation assays. The qPCR analysis revealed that 18S mRNA was predominantly localized in the cytoplasm, while U6 was primarily distributed in the nucleus of skeletal muscle satellite cells (SMSCs), confirming effective separation of nucleus and cytoplasm in SMSCs (Figure 5A,B). Importantly, during the proliferation phase of SMSCs, circ_14820 exhibited a distribution of 22.39% in the nucleus and 77.61% in the cytoplasm. Similarly, during the differentiation phase, circ_14820 was found to be 18.56% in the nucleus and 81.44% in the cytoplasm. These findings collectively indicate that circ_14820 predominantly localizes in the cytoplasm of SMSCs, underscoring its potential to function in post-transcriptional regulation.

### 2.6. Analysis of miRNAs Regulated by Circ_14820

To comprehensively investigate the downstream miRNAs influenced by circ_14820, total RNA was extracted from SMSCs overexpressing circ_14820 and their controls for small RNA sequencing. Analysis revealed 25 differentially expressed miRNAs (DEmiRNAs), with 10 showing up-regulation and 15 down-regulation (Figure 6A). A hierarchical clustering heatmap illustrates distinct clustering patterns among these DEmiRNAs (Figure 6B).

Further functional annotation of 327 target genes of the DEmiRNAs was conducted. This analysis identified significant enrichment across 64 KEGG signaling pathways and 268 gene ontology (GO) terms. Notably, pathways crucial to muscle development, such as adhesion plaque, Rap1 signaling, MAPK signaling, PI3K-AKT signaling, and FoxO signaling pathways, were prominently enriched (Figure 6C). Additionally, enriched GO terms encompassed functions like cAMP binding, cell migration, and transcriptional regulation (Figure 6D).

We selected 10 DEmiRNA for qPCR validation to determine that the small RNA-seq sequencing data were reliable (Figure 7).

### 2.7. Analysis of mRNAs Regulated by Circ_14820

We conducted a systematic exploration of mRNAs affected by circ_14820 through mRNA transcriptome sequencing, identifying 253 significantly differentially expressed mRNAs (DEmRNAs). Among these, 78 were up-regulated and 175 were down-regulated (Figure 8A,B). Functional enrichment analysis of these DEmRNAs revealed significant enrichment across 52 KEGG signaling pathways, including pivotal pathways such as the cell cycle, P53 signaling, and PI3K-AKT signaling pathways (Figure 8C). Additionally, among the 332 enriched gene ontology (GO) entries, critical terms such as myogenic differentiation, skeletal muscle cell differentiation, and cell division were prominently featured (Figure 8D). Furthermore, we constructed a protein–protein interaction (PPI) network of the DEmRNAs using the STRING online database. This network comprised 51 genes with interactions involving 139 pairs (confidence level > 0.9). Notably, we found that genes related to skeletal muscle development, such as BUB1, BUB1B, and CDK1, were identified as hub genes of the network (Figure 8E).

We randomly selected 10 DE mRNAs for qPCR validation and determined that the mRNA-seq data were reliable (Figure 9).

### 2.8. Analysis of Circ_14820-miR-206-CCND2 Regulatory Axis

According to bioinformatics analysis, potential binding sites between miR-206 and circ_14820 were identified (Figure 10A). Subsequently, we constructed a dual luciferase reporter vector based on these binding sites (Figure 10B). The results demonstrated that miR-206 mimics significantly reduced the relative luciferase activity of circ_14820-WT, whereas circ_14820-MT was unaffected (Figure 10C). This finding suggests that circ_14820 was able to target and regulate miR-206. Previous studies have shown that miR-206 promotes the proliferation of chicken adult myoblasts by targeting CCND2 [25]. In our study, we successfully modulated miR-206 expression levels (Figure 10D) and observed that miR-206 negatively regulated CCND2 expression. Importantly, circ_14820 partially attenuated the inhibitory effect of miR-206 on CCND2 expression (Figure 10E). Additionally, conservation analysis revealed high sequence similarity and a shared seed region (GGAAUGU) between miR-206 mature sequences across species such as goat and chicken (Figure 10F). Taken together, these results suggest that circ_14820 may alleviate the inhibitory effect of miR-206 on CCND2 expression through a sponge mechanism.

### 2.9. circ_14820 Regulates ATAD2 Expression through Novel miRNA 6-9154

In this study, we observed that the expression of the source gene ATAD2 was significantly upregulated following circ_14820 overexpression. Additionally, bioinformatics analysis suggested a potential targeting relationship between novel miRNA 6_9154 and ATAD2. Subsequently, we modulated the expression of novel miRNA 6_9154 (Figure 11A) and found that it negatively regulated ATAD2 expression. Interestingly, circ_14820 was able to reverse the inhibitory effect of novel miRNA 6_9154 on ATAD2 expression (Figure 11B). Further investigation revealed a potential binding site between circ_14820 and novel miRNA 6_9154 (Figure 11C), prompting us to hypothesize that circ_14820 might regulate ATAD2 expression through interaction with novel miRNA 6_9154. To validate this hypothesis, we constructed a dual luciferase reporter vector based on the binding site of circ_14820 and novel miRNA 6_9154 (Figure 11D). Unexpectedly, our experiments did not detect a direct binding relationship between circ_14820 and novel miRNA 6_9154 (Figure 11E). These results suggest that circ_14820 is unlikely to regulate ATAD2 expression by acting as a sponge for novel miRNA 6_9154.

### 2.10. Construction of Circ_14820-miRNA-mRNA Ternary Regulatory Network

In this study, we identified 15 potential circ_14820-miRNA relationship pairs through small RNA-seq analysis. Subsequently, we conducted an association analysis of miRNA and mRNA data, screening a total of 124 miRNA–mRNA relationship pairs, including the circ_14820 source gene ATAD2. Finally, we constructed a comprehensive circ_14820-miRNA-mRNA regulatory network relevant to goat skeletal muscle development (Figure 12).

## 3. Discussion

As circRNA research advances, their biological roles in organisms, including the regulation of skeletal muscle development, are becoming increasingly elucidated [26]. Previous studies have highlighted the spatiotemporal specificity of circRNA expression [9,27]. Additionally, circRNAs possess specific reverse splicing sites and exhibit resistance to degradation by nucleic acid exonucleases [28]. In our study, we identified that a novel circRNA, circ_14820, originating from exons 8 to 20 of the ATAD2 gene, contains a specific reverse splice site and resistance to RNase R, typical of exon-derived circRNAs. Notably, circ_14820 showed significantly higher expression in various tissues of 75-day-old embryonic goats, particularly enriched in the longissimus dorsi muscle. Moreover, overexpression of circ_14820 significantly promoted SMSC proliferation while inhibiting differentiation.

CircRNAs have a unique subcellular localization, and exon-type circRNAs are mainly localized in the cytoplasm, with excellent potential to exert post-transcriptional regulatory functions and are suitable to act as miRNA sponges [29]. For instance, circUSP13 releases IGF1 by targeting adsorption of miR-29c, thereby regulating differentiation and apoptosis of myoblasts [30]. Similarly, circUBE3A accelerates the proliferation and differentiation of adult goat myoblasts through the miR-28-5p-HADHB axis [31]. In this study, we determined that circ_14820 was mainly localized in the cytoplasm of SMSCs by nucleoplasmic separation techniques. Furthermore, bioinformatics techniques based on sequence complementarity and binding free energy to predict target miRNA molecules downstream of the target circRNA are not always confirmed by experimental results [32] and do not further reveal other potentially important novel miRNA molecules. Therefore, we used small RNA-seq and mRNA-seq to systematically reveal the circ_14820-miRNA-mRNA regulatory network.

Screening of differentially expressed miRNAs (DEmiRNAs) via small RNA-seq identified miR-1 and miR-206, prominent members of the MyomiRs family, with miR-206 specifically expressed in skeletal muscle [33]. As a pivotal regulator of skeletal muscle development, miR-206 facilitates myogenic differentiation by arresting cell cycle progression and promoting myotube formation [34]. Importantly, we confirmed the existence of a direct targeting relationship between circ_14820 and miR-206 by dual luciferase reporter assay, suggesting that circ_14820 may be involved in the regulation of goat skeletal muscle development by targeting adsorption of miR-206. Notably, in addition to myogenic miRNAs, other DEmiRNA molecules, such as miR-708-5p [35], miR-145-5p [36], miR-146b-5p [37], and miR-335-3p [38], have also been reported to be involved in the regulation of animal muscle development. Furthermore, functional annotation results indicated that the target genes of these DEmiRNAs were significantly enriched in typical myogenesis signaling pathways, including MAPK, FoxO, PI3K-AKT, and Rap1 [39,40,41,42].

Furthermore, our mRNA-seq analysis identified several DEmRNAs affected by circ_14820, including ATAD2, ACTC1, and CCND2. It has been reported that circRNAs converge with their parental genes in expression patterns [32,43]. Consistently, circ_14820 overexpression notably upregulated its parental gene ATAD2, suggesting a close association between the two. Notably, ACTC1 depletion impedes myotube formation and reduces the expression of key myogenic markers, such as MYOD1, MYOG, MYH3, MRF4, and MYF5, in adult myoblasts [44]. Meanwhile, CCND2, a downstream target of MyoD1, plays a pivotal role in cellular growth and metabolism through the PI3K-AKT pathway [45]. Thus, we hypothesize that circ_14820 regulates goat skeletal muscle development by downregulating ACTC1 and upregulating CCND2 expression.

Available studies have shown that HDAC4 is a conserved target gene of miR-206 [25,46,47]. Interestingly, previous studies have demonstrated that circ_RBFOX2 functions as a miR-206 sponge, leading to up-regulation of CCND2 expression while leaving HDAC4 unaffected [25]. Similarly, our findings reveal that miR-206 significantly suppresses CCND2 expression in goat SMSCs, and increased circ_14820 levels mitigate this inhibitory effect of miR-206 on CCND2 expression. However, no significant impact on HDAC4 expression was observed upon circ_14820 overexpression. Furthermore, circRNAs exert regulatory roles over parental genes through various mechanisms. For instance, EIcircRNA forms a complex with small ribonucleic U1 snRNP, subsequently binding to RNA polymerase II at the promoter region of its parental genes to modulate their expression [48]. Recent studies have also highlighted circRNAs’ ability to regulate parental genes through a competitive endogenous RNA (ceRNA) mechanism. For example, circ-STAT3 upregulates the expression of Gli2 and STAT3 by targeting miR-29a/b/c-3p [49]. In this study, we report a novel miRNA molecule, novel miRNA 6-9154, which negatively regulates the expression of the circ_14820-derived gene ATAD2. However, dual luciferase reporter assays did not confirm a direct binding interaction between circ_14820 and novel miRNA 6-9154. Therefore, it is improbable that circ_14820 acts as a sponge for novel miRNA 6-9154 to regulate ATAD2 expression.

## 4. Materials and Methods

### 4.1. Ethics Statement

The animal experiments involved in this study were approved by the Laboratory Animal Center of Southwest University under the approval number SWU_LAC-2023100138. All research methods were performed in strict accordance with the guidelines of the institute.

### 4.2. Animal

The animals used in this study were Dazu black goats from Southwest University Farm. The samples used for tissue expression profile construction were all obtained from one of our previous studies [50], including heart, liver, spleen, lung, kidney, stomach, pectoral muscle, hamstring muscle, and longissimus dorsi muscle of Dazu black sheep at 75 days of embryonic stage and 1 day after birth, and each sample contained three biological replicates.

### 4.3. Isolation and Identification of SMSCs

In this study, we obtained primary SMSCs isolated from the longissimus dorsi muscle of newborn goats with reference to the existing research method [18]. We used PAX7 [51] as the signature gene of SMSCs for their immunofluorescence identification, and the results of the identification are shown in Appendix A. Primary SMSCs were cultured in DMEM/F12 (Gibico, Guangzhou, China) containing 10% fetal calf serum (Gibico, Guangzhou, China) and 1% penicillin (Gibico, Guangzhou, China) in proliferation medium (GM). When SMSCs grew to 85% density, cells were digested using EDTA solution containing 0.25% trypsin (Solarbio, Beijing, China) and passaged or spread. When primary SMSCs were grown to 70% density, they were induced to initiate differentiation using differentiation medium (DM) containing 2% horse serum and 1% penicillin in DMEM/F12. All cells were cultured in a cell culture incubator at 37 °C and 5% CO_2_.

### 4.4. RNA Extractions and qPCR

Total RNA from tissues or cells was extracted using RNAiso Plus (TaKaRa, Akita, Japan) reagent according to the manufacturer’s specifications, and the concentration of RNA was determined using a NanoDrop 2000c spectrophotometer. Subsequently, RNA reverse transcription was performed using a PrimeScript™ RT kit or miRNA PrimeScript RT kit (TaKaRa, Kusatsu, Japan). TB Green Premix Ex Taq II premix (TaKaRa, Kusatsu, Japan) was added, and qPCR reactions were performed on a Bio-Rad CFX96 system (Bio-Rad, Shanghai, China). Each experiment contained at least three biological replicates and three technical replicates. We used GAPDH as an internal reference gene for mRNAs and circRNAs and U6 as an internal reference gene for miRNAs and calculated the relative gene expression using the 2^−ΔΔCT^ method [52]. The primers used in this study were all provided by Tianyi Huiyuan Biotechnology Co. (Wuhan, China), and the primer information is shown in Appendix A.

### 4.5. Circ_14820 Identification

Circ_14820 was derived from our previous high-throughput sequencing data (SRA database accession number: PRJNA749391). To confirm the circular structure of circ_14820, we validated circ_14820 at the circularization site by Sanger sequencing. Meanwhile, PCR reactions were performed on cDNA, RNase R (GENESEED, Guangzhou, China)-treated cDNA, or genomic DNA (gDNA) isolated from the longissimus dorsi muscle of goats using convergent primers and divergent primers. Finally, the bands of circ_14820 were shown by 1.2% agarose gel electrophoresis. In addition, we examined the tolerance of circ_14820 to nucleic acid exonucleases by qPCR, using the cDNA of the longissimus dorsi muscle of goats treated and untreated with RNase R as a template. The secondary structure information of circ_14820 was analyzed by the RNA fold web server (http://rna.tbi.univie.ac.at/cgi-bin/RNAWebSuite/RNAfold.cgi, accessed on 10 October 2023) for online prediction.

### 4.6. Vector Construction and Transfection

The circ_14820 overexpressing lentivirus used in this study was provided by Hanheng Biotechnology Co. Ltd. (Shanghai, China). The miRNA mimics (mimics), inhibitors (inhibtor), and their controls (mimics NC and inhibtor NC) were designed and synthesized by Sangon Biological Co (Shanghai, China).

According to the manufacturer’s instructions, the lentiviral infection multiplicity (MOI = 60) was used as the transfection condition (see Appendix A for the results of the screening of optimal MOI values), and lentiviral transfection was performed using the half-volume infection method when the SMSCs were grown to 30–50% confluence. Briefly, cells were first cultured using a half-volume of culture medium, and lentiviral transfection was performed, and after 4 h of transfection, the transfection was considered to be finished after replenishing the medium and continuing the culture for 24 h. The cells were then infected with lentiviruses using the half-volume infection method. Meanwhile, according to the manufacturer’s instructions, the riboFECTTMCP transfection kit (RiboBio, Guangzhou, China) was used in this study to transfect miRNA mimics or inhibitors and their controls into cells.

### 4.7. Cell Counting Kit-8 (CCK-8) and EdU Assay

Cell proliferation viability was detected using a Cell Counting Kit-8 Proliferation Kit (Solarbio, Beijing, China). Cells were inoculated in 96-well plates, and after cell transfection was completed, 10 μL of CCK8 reagent was added to each sample well. After that, the 96-well plates were incubated in an incubator at 37 °C, 5% CO_2_ for 1 h. Finally, the absorbance of all sample wells was measured at 450 nm.

EdU assay was performed using a 5-bromo-2-deoxyuracil (EDU) kit (RiboBio, Guangzhou, China), and the incubation, fixation, permeabilization, and cell staining were carried out strictly according to the instructions. Finally, images were captured using an inverted fluorescence microscope (OLYMPUS, Tokyo, Japan) and quantitative statistics were performed using ImageJ (version 2.0).

### 4.8. MyHC Immunofluorescence Assay

For MyHC immunofluorescence assay, cells were first fixed using 4% paraformaldehyde solution for 30 min at room temperature and treated with 0.5% Triton X-100 to make them permeable. After that, the cells were blocked using 10% goat serum and incubated overnight with goat anti-rabbit MyHC primary antibody (Proteintech, Wuhan, China). The cells were incubated with goat anti-rabbit IgG (H + L) secondary antibody (Proteintech, Wuhan, China) for 2 h away from light. After that, cells were stained for nuclei using 0.05 μg/mL DAPI nuclear staining reagent (Solarbio, Beijing, China). Finally, images were captured using an inverted fluorescence microscope (OLYMPUS, Japan) and quantitative statistics were performed using ImageJ (version 2.0).

### 4.9. Flow Cytometry Cell Cycle Analysis

The collected cells were resuspended using pre-cooled 75% ethanol at −20 °C and the cells were fixed at 4 °C for 24 h. Afterwards, cells were incubated using 500 mL of propidium iodide (PI)/RNase staining buffer (Coolaber, Beijing, China) at 37 °C for 15 min, followed by flow cytometric cycle analysis.

### 4.10. Western Blotting (WB) Assay

Total proteins were extracted from experimentally treated cells using a radioimmunoprecipitation kit (RIPA) (Beyotime, Shanghai, China). After that, protein concentration was determined using the BCA protein assay kit (Beyotime, China). Next, qualified protein samples (≥20 μg each) were individually loaded into polyacrylamide gel electrophoresis and transferred onto polyvinylidene difluoride (PVDF) membranes (Millipore, Burlington, MA, USA). The specific primary antibodies used were anti-PAX7 (dilution ratio of 1:500), anti-MyHC antibody (dilution ratio of 1:1000), anti-MyoD antibody (dilution ratio of 1:2000), and anti-ACTB (dilution ratio of 1:20,000) (Proteintech, Wuhan, China). Secondary antibodies used were goat anti-rabbit and goat anti-mouse IgG (H + L)-HRP coupling (Boster, Wuhan, China). The expression levels of target proteins were detected using the substrate chemiluminescence ECL method (Vazyme, Nanjing, China).

### 4.11. Nucleoplasmic Separation of SMSCs

To analyze the ratio of circ_14820 in the nucleus to cytoplasm of SMSCs, SMSCs were treated with a Cytosolic Nucleoplasmic Separation Kit (Invent, Eden Prairie, MN, USA), and total RNA was extracted from the nucleus and cytoplasm, respectively. after that, cDNAs were prepared according to the method of 2.4, and qPCR reactions were performed. The 18S mRNA and U6 served as the cytoplasmic and cytosolic endogenous reference genes, respectively [53]. The primer information is shown in Appendix A.

### 4.12. Small RNA and MRNA Sequencing Analysis

Library preparation and small RNA and mRNA sequencing were performed at AIQ Biotech (Wuhan, China), where the cell samples used for small RNA and mRNA sequencing were the same batch, i.e., total RNA samples extracted from the circ_14820 overexpression group and its control (SMSCs) (n = 3), respectively. The results of RNA quality testing and sequencing quality are shown in Appendix A. Quality test results and sequencing quality are shown in Appendix A.

In this study, RNA quality was examined by Qubit Fluoromete, Q-sep400 and agarose gel electrophoresis. Afterwards, cDNA libraries were constructed using NEBNext Ultra RNA Library Prep Kit and sequenced on the Illumina Nova Seq 6000 platform. The raw data were quality controlled using fastqc (version: 0.11.5) software to obtain clean reads.

Sequences of 18–30 bp in length were screened and annotated using the Rfam (14.1 http://rfam.xfam.org, accessed on 10 October 2023) database to remove non-miRNA sequences. Afterwards, clean small RNA was localized to the *Capra hircus* ARS1 reference genome using bowtie (v1.1.2) (https://ftp.ncbi.nlm.nih.gov/genomes/all/GCF/001/704/415/GCF_001704415.2_ARS1.2/GCF_001704415.2_ARS1.2_genomic.fna.gz, accessed on 10 October 2023). Novel miRNAs were predicted and identified based on the miRBase database, the signature hairpin structure of miRNA precursors, and miRDeep2 (v2.0.0.8). We used transcripts per million (TPM) (TPM = readCount × 10^6^)/total readCount) values to normalize miRNA expression.

The next step for clean reads of mRNAs was to localize them to the *Capra hircus* ARS1 reference genome using hisat2 software (version: 2.0.1-beta) and to count the number of reads landing on genes using featureCounts software (version: v1.6.0). We used FPKM (fragments per kilobase of exon per million reads mapped, FPKM = 10^9^ × C/NL) values to correct for mRNA expression, where C represents the number of fragments on a particular gene comparison; N represents the number of Fragments on all comparisons; and L represents the length of the gene.

Differentially expressed miRNAs (DEmiRNAs) and differentially expressed mRNAs (DEmRNAs) were screened by the DESeq2 tool in the R package, and the screening criteria for DEmRNA were *p*-adjust < 0.05 and |log2FoldChange| > 1; the screening criteria for DEmiRNA were FoldChange > 1.2.

Gene ontology (GO) analysis and Kyoto Encyclopedia of Genes and Genomes (KEGG) analysis were performed on KOBAS (http://bioinfo.org/kobas, accessed on 10 October 2023).

### 4.13. Analysis of the Luciferase Report

For the dual luciferase reporter gene assay, the circRNA sequence containing the target miRNA binding site, the full-length cDNA of the target miRNA -3′UTR and its corresponding mutant were cloned into the psi-Check2 vector to construct the wild-type plasmid (WT) and the mutant plasmid (MT), respectively. The miRNA mimics and their controls were transfected into 293T cells along with circ_14820-MT or circ_14820-WT. Cells were treated with the Dual-Luciferase Reporter Assay Kit assay kit (HANBIO, Shanghai, China) according to the manufacturer’s instructions, and subjected to fluorescent signal detection.

### 4.14. Bioinformatics Analysis

In this study, the protein–protein interactions (PPIs) network was constructed by STRING (https://cn.string-db.org, accessed on 10 October 2023). In addition, in order to construct the circ14820-miRNA-mRNA regulatory network, a collection of DEmiRNAs was screened with |log2FoldChange| > 0.4, and target gene prediction of DEmiRNAs was performed using Targetscan, mirDIP, and miRanda (v3.3a). In addition, based on the functional annotation information of DEmRNAs, mRNAs related to muscle development were screened and intersected with target genes of DEmiRNAs. After obtaining circ14820-miRNA and miRNA-mRNA relationship pairs, they were visualized by Cytoscape.

### 4.15. Statistical Analysis

Statistical analyses were performed using GraphPad Prism (version 6.02) software, and each experiment was conducted at least three times. Data were expressed as least squares mean ± standard error of the mean (SEM). The Student’s *t*-test was used for two-group comparisons, * *p* < 0.05, ** *p* < 0.01, and *** *p* < 0.001 considered statistically significant.

## 5. Conclusions

In this study, we identified a novel circular transcript circ_14820 from goat longissimus dorsi muscle tissue, which has an important effect on the proliferation and differentiation of goat SMSCs, elucidated the regulatory basis of its downstream miRNAs and mRNAs by small RNA-seq and mRNA-seq, and constructed a potential circ_14820-miRNA-mRNA regulatory network associated with skeletal muscle development in goats. Our results will help the understanding of the mechanism of circRNAs in muscle development and provide a theoretical reference for molecular breeding of goats.

## Figures and Tables

**Figure 1 ijms-25-08900-f001:**
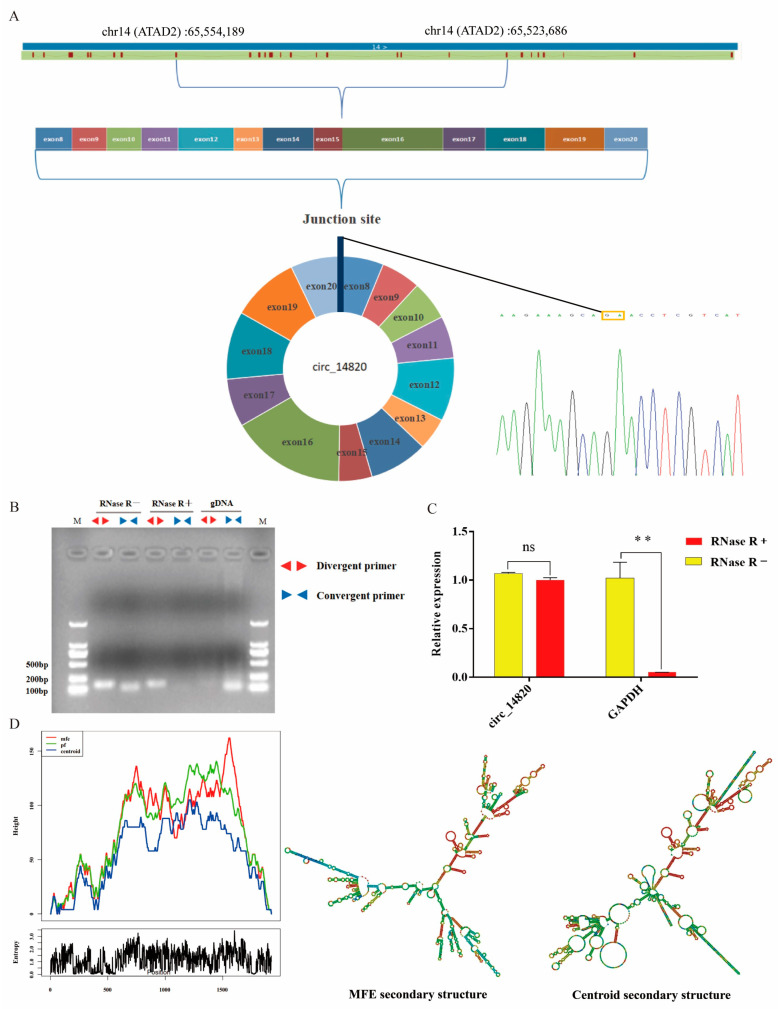
Identification of circ_14820. (**A**) Circ_14820 genomic location, exon composition, and shear site validation. (**B**) Validation of circ_14820 by convergent primers versus divergent primers. (**C**) Relative expression abundance of circ_14820 and GAPDH after RNase R treatment. (**D**) Secondary structure information of circ_14820. ** *p* < 0.01, ^ns^
*p* > 0.05.

**Figure 2 ijms-25-08900-f002:**
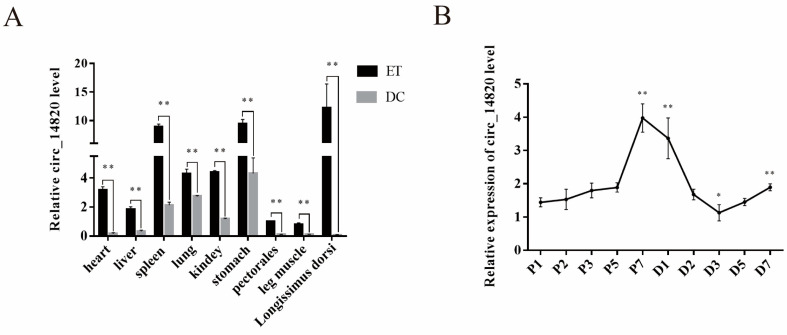
Circ_14820 expression profile. (**A**) Relative abundance of circ_14820 in goat heart, liver, spleen, lungs, kidneys, stomach, pectoralis, hamstrings, and longissimus dorsi muscles. (**B**) Expression trend of circ_14820 during proliferation and differentiation of SMSCs. Note: ET (embryonic 75-day-old DaZu black goats), DC (1-day-old postnatal DaZu black goats). * *p* < 0.05, ** *p* < 0.01.

**Figure 3 ijms-25-08900-f003:**
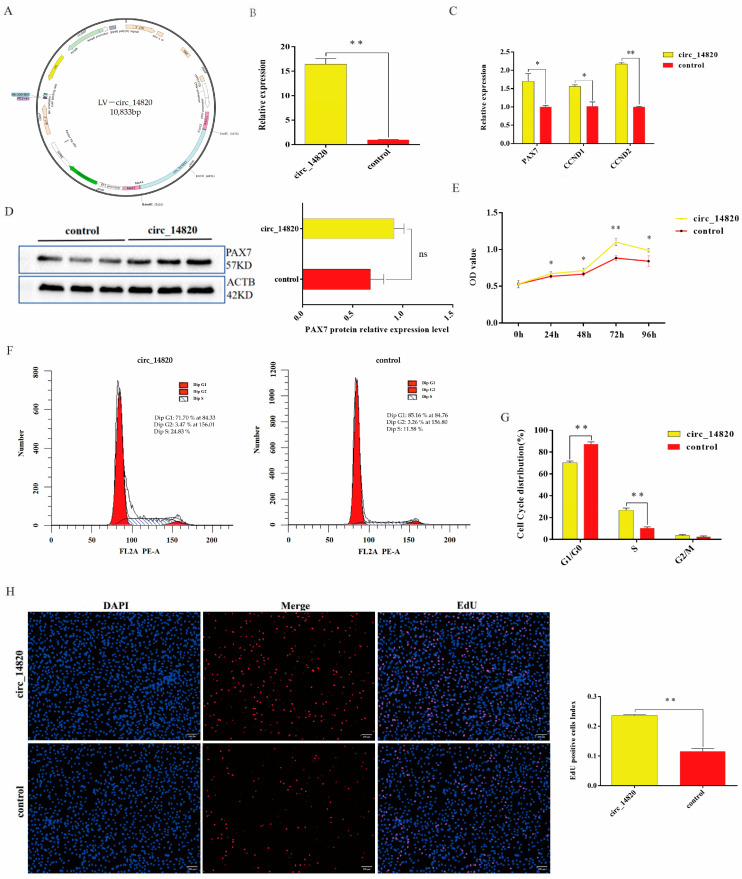
Effect of overexpression of circ_14820 on the proliferation of SMSCs. (**A**) Schematic diagram of circ_14820 overexpression vector construction. (**B**) Overexpression efficiency assay of circ_14820. (**C**) Effect of circ_14820 on the expression of SMSC proliferation marker genes. (**D**) Effect of circ_14820 on the protein expression of the proliferation marker gene PAX7 in SMSCs. (**E**) Absorbance of cells at 450 nm. (**F**,**G**) Effect of circ_14 820 on the cycle of SMSCs. (**H**) Statistical results of the number of EdU-positive nuclei and the proportion of EdU-positive cells in the cells. * *p* < 0.05, ** *p* < 0.01, ^ns^
*p* > 0.05. Scale bar: 200 μm.

**Figure 4 ijms-25-08900-f004:**
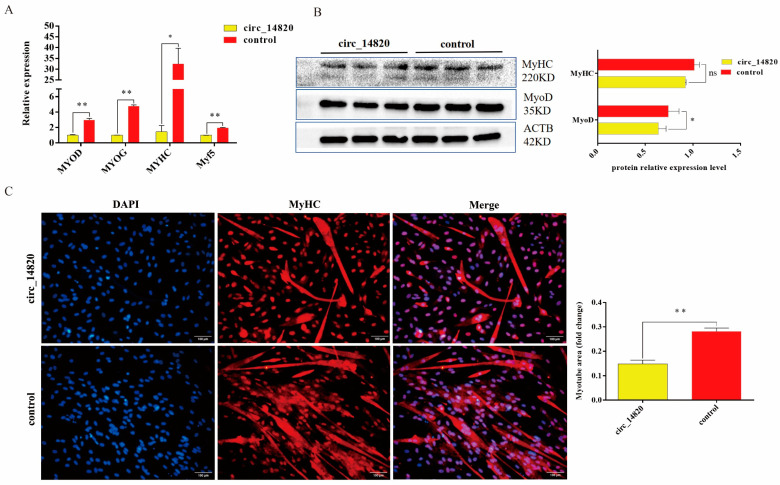
(**A**) Effect of circ_14820 on the expression of differentiation marker genes of SMSCs detected by qPCR. (**B**) Effect of circ_14820 on the protein expression of MyHC and MyoD, the differentiation marker genes of SMSCs. (**C**) Effects of circ_14820 on myotube production and statistical results of myogenic index detected by immunofluorescence. * *p* < 0.05, ** *p* < 0.01, ^ns^
*p* > 0.05.

**Figure 5 ijms-25-08900-f005:**
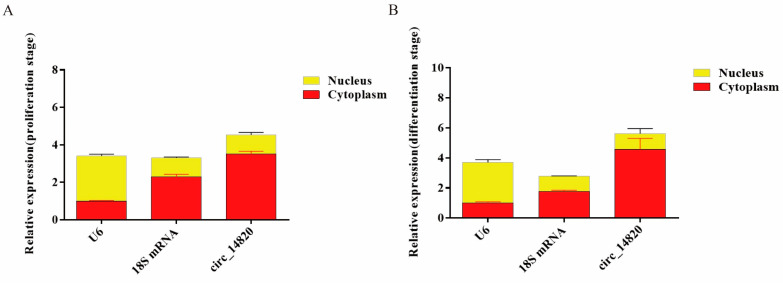
The circ_14820 ratio in the nucleus to cytoplasm of SMSCs. (**A**) Nucleoplasmic localization of circ_14820 in the proliferation stage of SMSCs. (**B**) Nucleoplasmic localization of circ_14820 in the differentiation stage of SMSCs.

**Figure 6 ijms-25-08900-f006:**
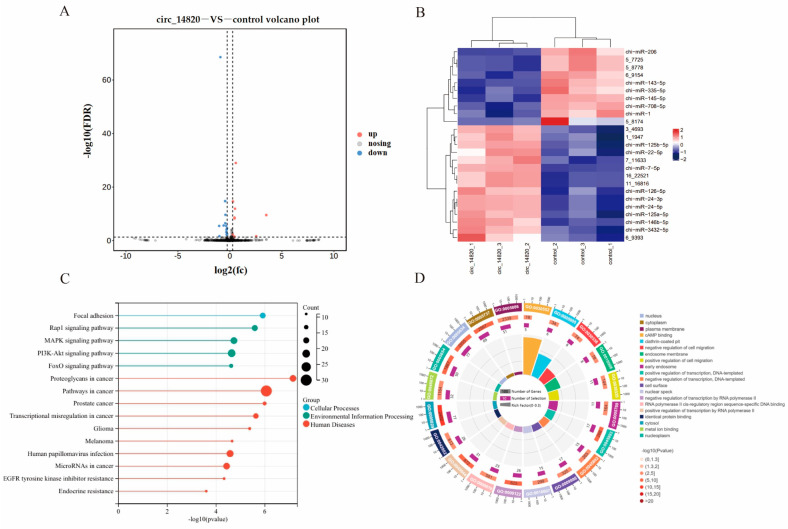
The miRNA analysis of circ_14820 regulation. (**A**) Volcano map of differentially expressed miRNAs. (**B**) Clustering heat map of differentially expressed miRNAs. (**C**) KEGG enrichment analysis of target genes of differentially expressed miRNAs. (**D**) GO enrichment analysis of target genes of differentially expressed miRNAs.

**Figure 7 ijms-25-08900-f007:**
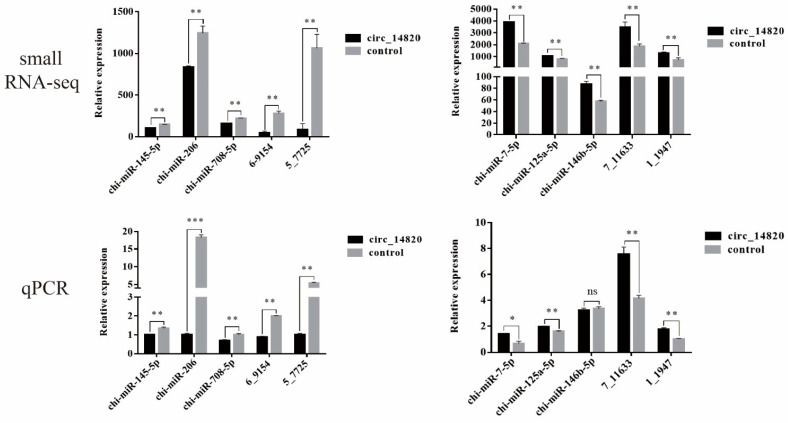
Validation of differentially expressed miRNA data. * *p* < 0.05, ** *p* < 0.01, *** *p* < 0.001, ^ns^
*p* > 0.05.

**Figure 8 ijms-25-08900-f008:**
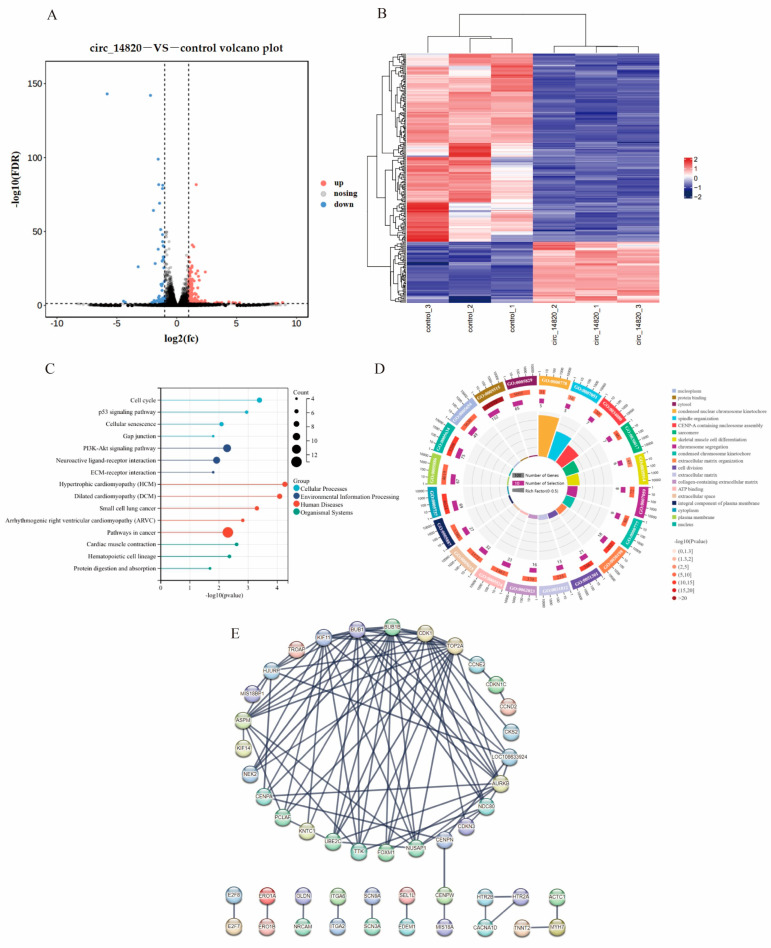
The mRNA analysis of circ_14820 regulation. (**A**) Volcano mapping of differentially expressed mRNAs. (**B**) Clustering heat map of differentially expressed mRNAs. (**C**) KEGG enrichment analysis of differentially expressed mRNAs. (**D**) GO enrichment analysis of differentially expressed mRNAs. (**E**) PPI network of DEmRNAs.

**Figure 9 ijms-25-08900-f009:**
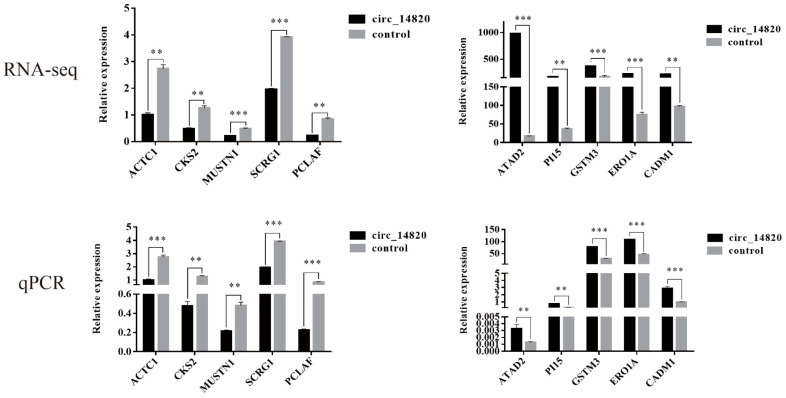
Validation of differentially expressed mRNA data. ** *p* < 0.01, *** *p* < 0.001.

**Figure 10 ijms-25-08900-f010:**
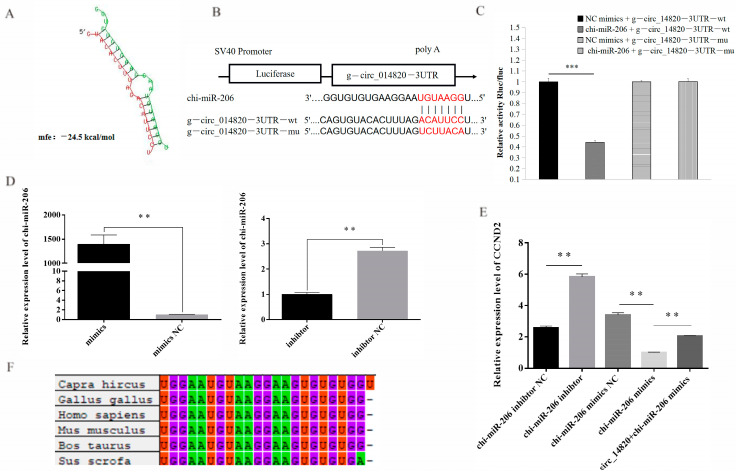
The circ_14820-miR-206-CCND2 regulatory axis analysis. (**A**) Predicted binding sites of circ_14820 to miR-206 using RNAhybrid software. (**B**) Schematic diagram of circ_14820 wild-type and mutant luciferase vector construction. (**C**) Dual luciferase activity assay. (**D**) The miR-206 overexpression and interference efficiency assay. (**E**) Effects of circ_14820 and miR-206 on CCND2 expression. (**F**) Sequence conservation analysis of miR-206 in goat, chicken, human, mouse, cow, and pig. ** *p* < 0.01, *** *p* < 0.001.

**Figure 11 ijms-25-08900-f011:**
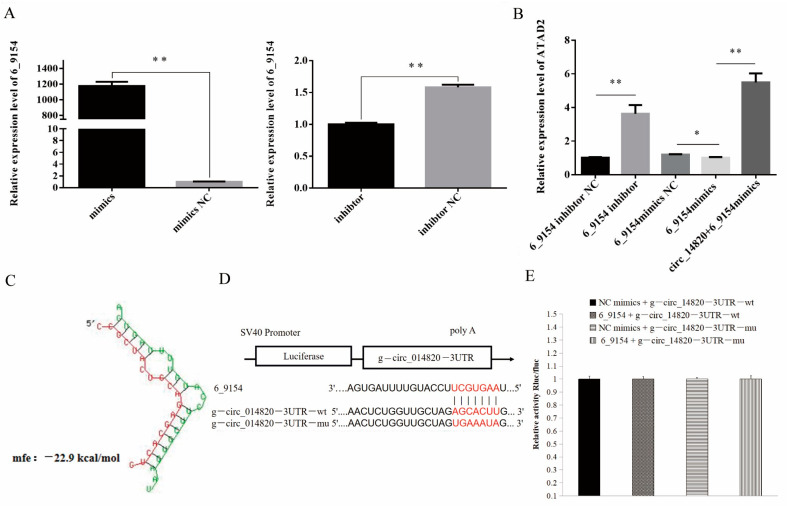
(**A**) Novel miRNA 6_9154 overexpression and interference efficiency assay. (**B**) Effect of circ_14820 and novel miRNA 6_9154 on CCND2 expression. (**C**) Binding sites of circ_14820 and novel miRNA 6_9154 were predicted using RNAhybrid software. (**D**) Schematic diagram of circ_14820 wild type and mutant luciferase vector construction. (**E**) Dual luciferase activity assay. * *p* < 0.05, ** *p* < 0.01.

**Figure 12 ijms-25-08900-f012:**
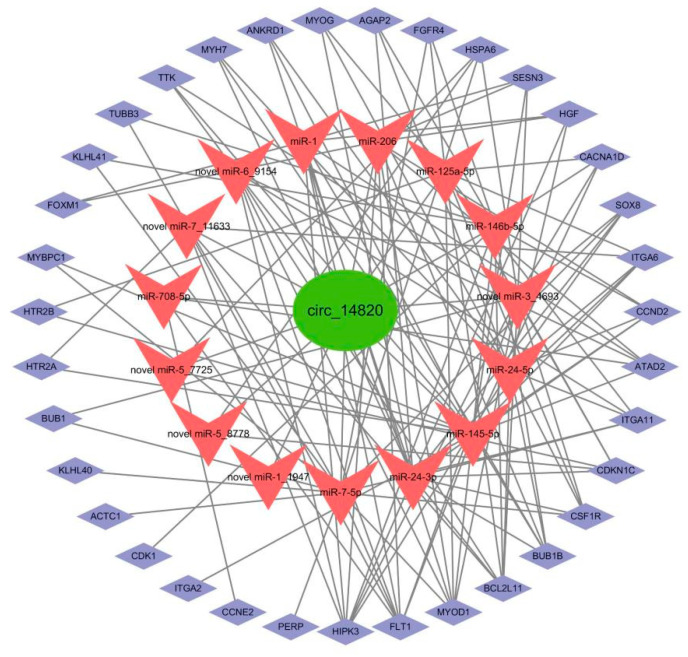
The circ_14820-miRNA-mRNA regulatory network.

## Data Availability

Datasets generated/analyzed during the current study are available. Raw sequencing data are available via NCBI data accession number PRJNA1104785.

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
