# Peer review of "Role and Regulatory Mechanism of circRNA_14820 in the Proliferation and Differentiation of Goat Skeletal Muscle Satellite Cells"

_ijms, 2024, doi:10.3390/ijms25168900_

Round 1
Reviewer 1 Report
Comments and Suggestions for Authors
The authors present an interesting study where they have investigated the role of the circular RNA, circRNA_14820, in the skeletal muscle satellite cell (SMSC). After identifying circRNA_14820 they have investigated its role in the development of SMSCs.
The introductory text provides sufficient background information to enable a good understanding and the underpinning rationale for the following study.
The materials and methods are well described and would enable the replication of the study in SMSC or other cells.
In general, the results are well presented and described. The results also support the conclusions drawn by the authors. The resolution of some of the figures and their size made it difficult to see the details in some panels. I guess expanded versions if the manuscript is published would help solve this issue.
While I commend the authors for providing the original Western blots as supplemental files, the blots should be provided as full size of the blotting membrane, ie not cropped. A key purpose of providing these images is to allow the reader to see the whole Western experiment. These would for example allow the reader to determine the degree of non-specific binding to the membrane.
Note is the Fig 3D for PAX7 inverted in the supplemental file? The lighter bands appear to have switched sides.
Line 3 It would be better not to have an abbreviation in the title.
Line 16 suggest revision “Our preliminary findings suggest that”
Line 20 suggest adding "Circ_14820" and "mir-206" to the keywords.
Line 88 Figure 1 – It is difficult to see the details in the diagrams of Fig 1A.
Line 104 Figure 2 – I would suggest explaining the abbreviations “ET” and “DC” in the legend. While they are explained in the main text, the figure should be interpretable independently of the text.
Author Response
Dear reviewers.
I would like to express my appreciation for your efforts and hard work. Your expertise and rigour have given me a great deal of respect for your work. I have benefited from your detailed and constructive review and advice on my manuscript. You are not just a reviewer, but a dedicated and responsible expert who has made a significant contribution to the advancement of academia. Once again, I thank you for all your efforts on my manuscript and look forward to receiving your guidance and assistance again in the future. I wish you all the best in your endeavours and good health!

Reviewer 2 Report
Comments and Suggestions for Authors
This manuscript presents preliminary results about the effect of circRNA_14820 on the proliferation/differentiation of goat skeletal muscle satellite cells. A novel circular transcript circ_14820 from goat longissimus dorsi [italic] muscle tissues was identified.
Title – adapt to read as: Role and regulatory mechanism of circRNA_14820 in the proliferation and differentiation of goat skeletal muscle satellite cells
The main question addressed by this study is circ_14820, an exonic transcript derived from adenosine triphosphatase family protein 2 (ATAD2).
Parts considered relevant for the field are: (a) the identification of a novel circular transcript circ_14820 from goat longissimus dorsi muscle tissues; (b) its important effect on the proliferation and differentiation of goat skeletal muscle satellite cells (SMSC); and (c) and elucidated the regulatory basis of its downstream miRNAs and mRNAs by small RNA-seq and mRNA-seq. In fact, the overexpression of circ_14820 markedly enhanced the proliferation of goat SMSC while concomitantly suppressing their differentiation.
Findings preliminarily suggest that the circ_14820-miR-206-CCND2 regulatory axis may govern the development of goat SMSC. These discoveries contribute to a deeper understanding of circRNA-mediated mechanisms in regulating skeletal muscle development, thereby advancing the knowledge on muscle biology.
Dazu Black goats represent a Chinese indigenous breed, a circumstance that may bias comparison with other goat breeds.
The conclusions seem to be consistent with the presented arguments. The main questions were addressed, specifically by isolation and identification of SMSC, RNA extraction and qPCR, circ_14820 identification, vector construction and transfection, among several other related procedures
Line 7 – adapt to read as (plural): Skeletal muscle satellite cells (SMSC), a type of myogenic stem cells, play a pivotal role in
Line 12 – replace Significantly with Importantly
Keywords – display alphabetically
Line 26 – use MRF (without “s”) – change accordingly throughout the manuscript
Line 56, etc.- write longissimus dorsi in italics
Author Contributions – use initials for the names of authors
There are 53 references, which seem to be appropriate. Almost 805 of these references have been published in the last 10 years.
The manuscript is well illustrated by 12 figures, being all of them multiple or composite figures.
Author Response
Dear reviewers,I would like to express my special thanks for your hard work and patience with my manuscript. You have thoroughly read and carefully reviewed the content of my research and provided valuable suggestions and comments. These valuable suggestions are of great significance to my research work and will undoubtedly enrich and diversify my study. I greatly admire and appreciate your professionalism and meticulous review work. Your hard work has not only played a key role in my research, but has also given me a deep appreciation of the warmth of mutual support and assistance from my academic peers. Once again, I would like to thank you for your dedication and hard work and wish you all the best in your work, health and reading!
